Subject Areas:
environmental science, ecology, plant science

Keywords:
fire, forest degradation, biomass, demography, growth, morphological traits

Author for correspondence:
Aline Pontes-Lopes
e-mail: aline.lopes@inpe.br, alineplopes@gmail.com

# Drought-driven wildfire impacts on structure and dynamics in a wet Central Amazonian forest

Aline Pontes-Lopes[1], Camila V. J. Silva[2,3], Jos Barlow[2], Lorena M. Rincón[4], Wesley A. Campanharo[1], Cássio A. Nunes[5], Catherine T. de Almeida[1,6], Celso H. L. Silva Júnior[1,7], Henrique L. G. Cassol[1], Ricardo Dalagnol[1], Scott C. Stark[8], Paulo M. L. A. Graça[4] and Luiz E. O. C. Aragão[1,9]

[1]Earth Observation and Geoinformatics Division, National Institute for Space Research (INPE), São José dos Campos 12227-010, Brazil
[2]Lancaster Environment Centre, Lancaster University, Lancaster LA1 4YQ, UK
[3]Amazon Environmental Research Institute (IPAM), Brasília 71503-505, Brazil
[4]National Institute for Research in Amazonia (INPA), Manaus 69067-375, Brazil
[5]Department of Ecology and Conservation, Federal University of Lavras (UFLA), Lavras 37200-000, Brazil
[6]Department of Forest Sciences, Luiz de Queiroz College of Agriculture, University of São Paulo (USP/ESALQ), Piracicaba 13418-900, Brazil
[7]Department of Agricultural Engineering, State University of Maranhão (UEMA), São Luís 65055-310, Brazil
[8]Department of Forestry, Michigan State University, East Lansing, MI 48824, USA
[9]College of Life and Environmental Sciences, University of Exeter, Exeter EX4 4RJ, UK

AP-L, 0000-0001-7668-1226; CVJS, 0000-0002-4867-9871; CHLSJ, 0000-0002-1052-5551

While the climate and human-induced forest degradation is increasing in the Amazon, fire impacts on forest dynamics remain understudied in the wetter regions of the basin, which are susceptible to large wildfires only during extreme droughts. To address this gap, we installed burned and unburned plots immediately after a wildfire in the northern Purus-Madeira (Central Amazon) during the 2015 El-Niño. We measured all individuals with diameter of 10 cm or more at breast height and conducted recensuses to track the demographic drivers of biomass change over 3 years. We also assessed how stem-level growth and mortality were influenced by fire intensity (proxied by char height) and tree morphological traits (size and wood density). Overall, the burned forest lost 27.3% of stem density and 12.8% of biomass, concentrated in small and medium trees. Mortality drove these losses in the first 2 years and recruitment decreased in the third year. The fire increased growth in lower wood density and larger sized trees, while char height had transitory strong effects increasing tree mortality. Our findings suggest that fire impacts are weaker in the wetter Amazon. Here, trees of greater sizes and higher wood densities may confer a margin of fire resistance; however, this may not extend to higher intensity fires arising from climate change.

## 1. Introduction

Widespread occurrence of forest fires in the Amazon have become recurrent in recent years, driven by more frequent and intense droughts [1–3], increasing ignition sources associated with deforestation [4,5], and widespread forest degradation and fragmentation [6,7]. Wildfires are thus one of the key factors determining the likelihood of large-scale forest dieback in the Amazon [8]. Fire impacts on forests are amplified by drought [9], leading to long-term changes in forest structure and biomass [10,11]. Forest fires burned an estimated 9246 km² in the Brazilian Amazon, approximately 24% of all fires, during the

extreme El Niño drought of 2015/2016, a share of fire area that was two times greater than during the previous extreme drought (2010) [3]. The 2015/2016 drought strongly impacted even the Central Amazon, a moist forest region with widespread intact forest, causing the highest surface temperature and rainfall deficit anomalies ever registered in the region [3,12].

Studies from regions in the Amazon with a marked rainfall seasonality—which have more than three months of dry season (consecutive months with average rainfall lower than 100 mm month$^{-1}$) [13]—have made important progress in understanding how forest biomass stocks and dynamic processes (e.g. tree growth, recruitment and mortality) change after a fire event. In these more seasonal regions, fire implies marked changes in the forest structure: immediate near-complete mortality of saplings [14]; death of 36–74% of trees with diameter of 10 cm or more at breast height (DBH) up to 3 years after fire [14–18] and a delayed increase in mortality of larger trees (DBH $\geq 50$ cm) [19]. These structural changes lead to 49% loss in live aboveground tree biomass within 3 years after fire [19]. Delayed losses in biomass were estimated to occur from 5 to 8 years after the fire and persist for at least three decades, resulting from the mortality of large and high-wood density (WD) trees and insufficient regrowth to compensate losses [20]. This fire legacy is affected by the local/regional species composition and related morphological traits (e.g. plant size, WD, bark thickness and the presence of buttress roots) [21–24]. Another important predictor of tree mortality is fire intensity [21,22], which is enhanced by severe droughts [9].

Less seasonal regions in the Amazon—which have 3 months or less of the dry season—are dominated by undisturbed dense-closed tropical forests [25]. These forests maintain a moist microclimate even during the dry season peak, reducing or preventing fire spread even following prolonged dry periods [26]. Severe drought conditions, however, increase deposition of leaf litter and woody debris (surface fuel) [27] and lower moisture content, supporting more intense forest fires with longer duration and faster understorey spread [9]. Therefore, the existing gradients of rainfall seasonality and dry season length over the Amazon [13] may imply a gradient of climate-induced fuel moisture as well, limiting the occurrence of large forest fires over the biome in years of regular rainfall [28]. The less seasonal regions of the Central Amazon are, thus, near the end of this gradient, receiving high levels of rainfall and having forest fires restricted to severe drought years, as reported for 1998/1999, 2009/2010 and 2015/2016 [19,24,29].

To expand knowledge of fire impacts on Amazonian forests, we aimed to address three important gaps with the present study. The first knowledge gap is spatial: there is limited information from less seasonal regions (figure 1a), including critical areas under threat of forest degradation such as the Purus-Madeira interfluve, which links the central and southeastern Amazon [34]. The second gap is temporal: few studies show temporal trajectories of changes in forest functioning after fire [9,20,24] despite this being crucial to quantify impacts with less bias [35]. The third gap is in understanding to what extent plant morphological traits can avoid the effects of fire intensity on post-fire mortality [21–23] and post-fire recovery [24].

To address these knowledge gaps, we installed permanent plots in the northern Purus-Madeira immediately after an uncontrolled forest fire (approx. two months) that occurred during the 2015/2016 El Niño. We monitored these plots annually for more than 3 years (2015–2018). Employing this unique dataset in the Purus-Madeira region, we studied the post-fire forest dynamics and stem sensitivity to fire by addressing the following questions. (Q1) Fire behaviour: what was the fire intensity and coverage? (Q2) Structure: what was the magnitude of post-fire changes in total stem density and aboveground biomass (AGB), and which growth forms (trees, palms and lianas) and tree sizes contributed the most to these changes? (Q3) Demographics: which processes (mortality, growth and recruitment) drove the post-fire AGB dynamics? (Q4) Growth drivers: how did fire intensity and stem-level morphological attributes influence the post-fire growth of surviving trees? (Q5) Mortality drivers: to what extent did fire intensity and stem-level morphological attributes predict tree and palm mortality after the fire, and how did the relative importance of these predictors change over time?

## 2. Methods

### (a) Study area

The study area is located at about 90 km southeast of Manaus (Brazil), in the north portion of the Purus-Madeira interfluve in the municipality of Autazes (near the BR-319 road), in the Central Amazon (figure 1b). About 10% of the regional land cover is dairy cattle pasture, the primary agricultural activity in the northern interfluve [30,36]. The interfluve is an extensive flat region dissected by broad rivers, with subtle soil and topographic gradients, and medium to low soil drainage [37]. The soils are described as shallow Haplic Plinthosols [38]. The northern Purus-Madeira is covered by dense lowland non-flooded (terra-firme) and seasonally flooded forests [39]. The regional average annual rainfall ranges from 2000 to 2400 mm [33]. The dry season typically ranges from one to three months in length (figure 1a), including July to September.

The northern interfluve was markedly affected by the extreme 2015/2016 drought, as indicated by the lowest maximum cumulative water deficit (MCWD) in the 1990–2018 interval (figure 1c). This drought event was stronger than the droughts of 1982/1983 and 1997/1998, which were also related to El Niño events [12]. During 2015, the area burned by forest fires increased approximately 12 times in the northern interfluve in comparison to the 2001–2018 average (excluding 2015), affecting approximately 230 km$^2$. Contrastingly, there were lower forest fire extents in the central Purus-Madeira region (figure 1b), which corresponded with the greatest coverage of conservation units and the unpaved portion of the BR-319.

### (b) Data

Eighteen permanent inventory plots (0.25 ha each) were installed in the study area (figure 1d) in December 2015. Plots were located in 'terra-firme' forests where 12 plots were in burned forests and 6 were in unburned forests. All plots were installed about 2 months after the forest fires which were found by visual inspection of Landsat images and personal communication with local residents during preliminary field surveys (electronic supplementary material, figure S1). All plots were located in private lands, with permission from landowners for installation and continued monitoring. During the first measurement of burned plots, the forest understorey was open because of clear fire impacts with the forest floor displaying large ash and burned wood debris accumulation. There was a combination of burned and unburned forest

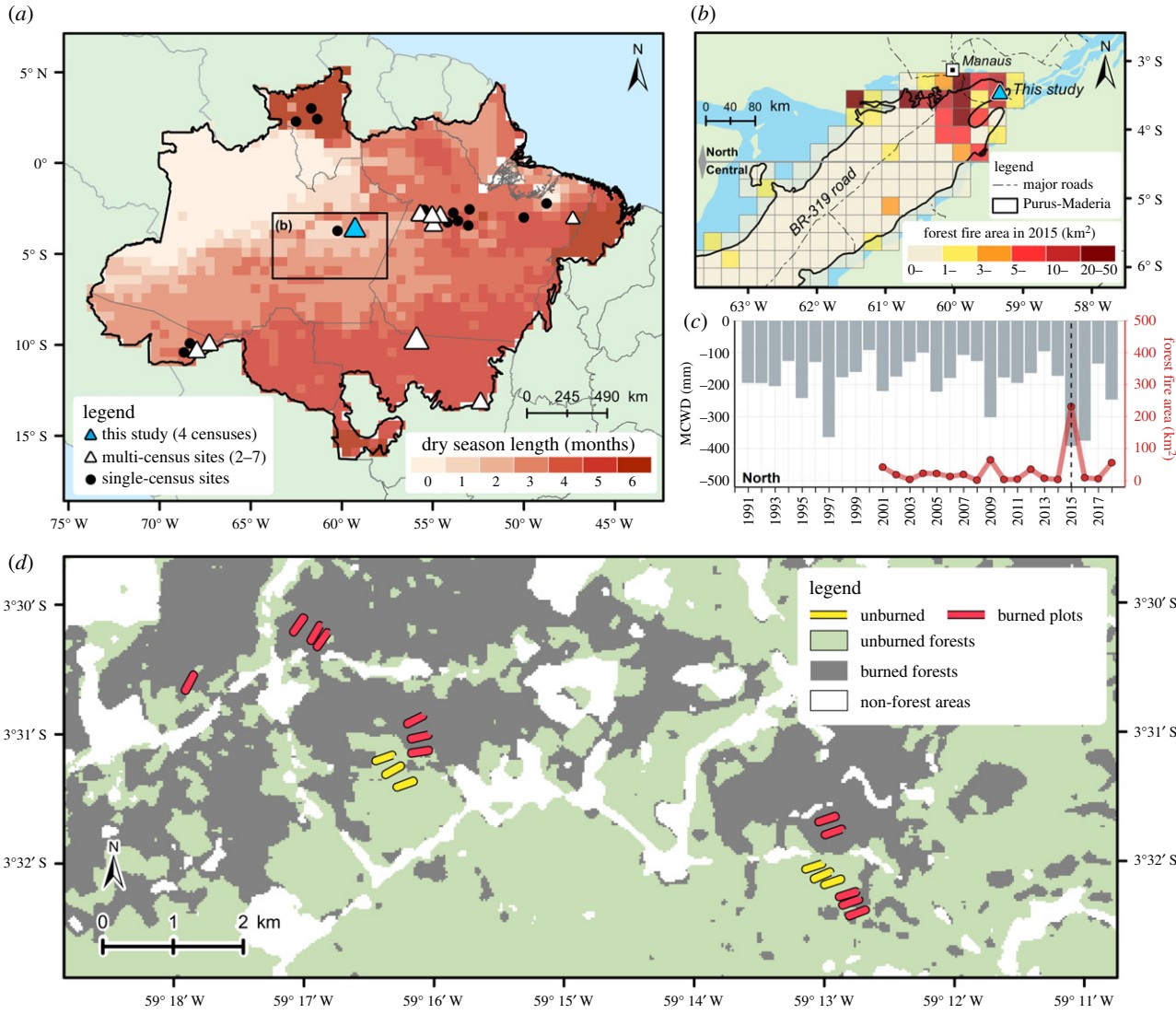

**Figure 1.** Contextualization of the study area within the Brazilian Amazon and the Purus-Madeira moist forest ecoregion. (*a*) Rainfall seasonality in the biome [13] and the location of this and other published studies with inventory plots in burned forests. (*b*) Gridded map showing the 2015 total burned forest area in part of the ecoregion [30–32]. (*c*) Yearly burned area (dots) and MCWD (bars) [33]. The dashed line indicates when our field measurements started (2015). (*d*) Local perspective of our study area, indicated by a blue triangle in (*a,b*). The 2015 forest fires are mapped in dark grey. Further details for this figure are in the electronic supplementary material, table S1 and text S1. (Online version in colour.)

patches within the burned area. We were able to set some of our plots during the wildfires. However, this was not a burn experiment, and we did not set and control the fire. Furthermore, we consider that the surveyed burned forest areas, composed by natural and human-made borders (figure 1*d*), is regionally representative once wildfires do not occur naturally in this wet environment, depending on ignition vectors associated with human presence and climatological conditions [40].

Each plot was a transect of 250 × 10 m, divided into 10 sectors. All plots were remeasured every November from 2016 to 2018 (see photos from these field campaigns in the electronic supplementary material, figure S2), totalling four censuses and three intervals of measurements (2015 to 2018). Despite not randomly placed, these long rectangular plots minimize potential selection bias [15]. In these plots, we registered all live trees (*n* = 2420), palms (200) and lianas (33) with DBH ≥ 10 cm following the Rainfor protocol [41]. We noted their taxonomic information and used flexible metric tapes to measure their stem circumferences at 1.3 m height (or above basal irregularities), values which were then converted into DBH. As a proxy for fire intensity [9,15,24], we recorded the char height (CH) of all burned trees and palms in the first census—determined as the highest clear mark of charring on the stem base. Mean WD

was retrieved for each individual from the Global Wood Density Database [42], according to its most detailed taxonomic identification [43], being 42% of all individuals classified at the species level, 50% up to the genus level only and 2% at the family level only. The remaining 6% were not identified and received plot-level mean WD values.

We used this dataset to calculate the stem density and AGB for each plot in each census. AGB was calculated using one biomass equation for each measured growth form: [43] for trees; [44] for palms and [45] for lianas. The biomass equations for trees and palms used height values obtained from local height–diameter regression models (electronic supplementary material, text S2).

## (c) Data analyses

To estimate the fire coverage and intensity (addressing Q1), we calculated, respectively, the number of fire-affected stems per plot and the mean CH on these individuals per plot and sector.

To assess the magnitude of post-fire structural changes (Q2), we calculated the temporal changes (Δ%) in stem density and AGB stocks, relative to the initial estimates:

$$S\Delta\%_{c,p,y} = \frac{(S_{c,p,y} - S_{c,p,2015})}{S_{c,p,2015}} \times 100,$$

where $S$ = structural metric of interest (stem density or AGB); $c$ = forest class (burned or unburned); $y$ = year of each census (2015 to 2018) and $p$ = each plot. Therefore, we assumed that the 2015 forest structure and biomass stocks were similar to pre-disturbance conditions, because we do not expect considerable biomass losses within the two months between the fire and our surveys. Then, we calculated and plotted the mean and standard deviation between plots. This approach was also used to calculate the temporal changes of both metrics in DBH size classes, considering trees only.

Forest dynamics were analysed by calculating the following annualized rates for each plot: growth, recruitment, mortality, wood productivity (WP) and net biomass change (net AGBΔ). We considered recruitment as the AGB of those living individuals which reached the minimum DBH (10 cm) after the previous census. WP corresponds to the recruitment added to the growth of all living individuals included in the previous measurement. Net AGBΔ is the net of WP gains and mortality losses. To identify which of these metrics drove the post-fire dynamics (Q3), we normalized each annualized rate by the AGB at the beginning of each interval:

$$D\Delta\%_{c,p,y} = \frac{D_{c,p,y}}{AGB_{c,p,y-1}} \times 100,$$

where $D$ = dynamics' metric of interest (growth, recruitment, mortality, WP or net AGBΔ); $c$ = forest class (burned or unburned); $y$ = year of each census (2016 to 2018) and $p$ = each plot. Finally, we used Mann–Whitney $U$-tests to assess whether there were differences in each metric's mean between unburned and burned plots in each interval.

To identify factors driving post-fire growth at tree stem-level (Q4), we matched trees from burned and unburned plots and ran generalized linear mixed models (glmm). First, we calculated radial growth and carbon accumulation, which are respectively the increment in DBH and the increment in biomass multiplied by a carbon content factor of 50% for each stem. For this analysis, we considered the entire 3-year interval only, instead of independent 1-year intervals—the longer interval ensures less noise in growth. We then paired surviving trees of similar DBH and WD between the burned and unburned plots following this procedure [24]: we randomly selected a burned plot stem first and then selected from among unburned plot stems that were within 10% margins of both DBH and WD of the burned plot reference stem, selecting the tree with the closest DBH. We thus created 575 pairs and calculated their differences in radial growth and carbon accumulation. We then applied glmm (with a normal error structure) to predict these differences, with three continuous predictor variables (CH, DBH and WD). We standardized all predictors (0–1) and set the burned plots as a random effect. Finally, we compared each variable's effects on the difference in tree growth and plotted the estimates of the adjusted models.

We have also used glmm to study the factors affecting post-fire mortality (Q5). The same predictor variables (CH, DBH and WD) were used to predict stem mortality probability in burned plots, using a binomial error structure (logistic glmm), and setting plots as a random effect. Analyses were performed separately for trees and palms in each 1-year interval, using all stems alive at the beginning of each interval. We ranked all model combinations according to their delta corrected Akaike information criterion (ΔAICc). We then repeated this analysis for unburned and burned plots, using only DBH and WD as predictors, to assess the differences in model selection according to fire occurrence. We also quantified the predicted marginal effects of each variable from the full model—by computing model predictions varying one predictor variable of interest while holding the other predictors constant at their averages. Finally, we rescaled the variables to evaluate effect size. After rescaling,

each variable's 1-unit represented 30 cm for CH, 15 cm for DBH and 0.1 g cm$^{-3}$ for WD. We then compared each predictor's effects on tree mortality by computing the model estimates (odds ratio).

All analyses were performed in R v. 4.0.2 using the packages: lme4, sjplot, MuMIn and ggeffects [46–49].

# 3. Results

## (a) Fire behaviour

Plots' fire coverage was on average $70 \pm 17\%$ with values ranging from 40 to 92% with two plots having less than 50% burn coverage. Burnt stems had average CHs of $27 \pm 4$ cm, with most trees burned on the stem base, up to 30 cm height (78% of all burnt stems). The burn patterns within each plot are exposed in electronic supplementary material, figure S3.

## (b) Changes in stem density and biomass stocks

Over the 3 years after the fire, stem density decreased from $517.7 \pm 38.8$ to $376.0 \pm 53.2$ stems ha$^{-1}$ while AGB decreased from $223.6 \pm 66.7$ to $193.7 \pm 49.7$ Mg ha$^{-1}$ in the burned plots (electronic supplementary material, table S2). These values represented losses of $27.3 \pm 9.0\%$ in stem density and $12.7 \pm 9.1\%$ in AGB (figure 2a,b). While the AGB of unburned plots remained steady over the studied period (approx. $182 \pm 40$ Mg ha$^{-1}$), the total stem density slightly decreased $6.0 \pm 5.0\%$ (from $518.4 \pm 51.9$ to $485.6 \pm 34.0$ stems ha$^{-1}$).

Among all growth forms, trees made up the largest losses in stem density (94.8%) and AGB (85.0%) in the burned plots. Note that trees were 94.0% of the overall AGB in 2015. These losses, in both stem density and AGB, were concentrated within small to medium-sized trees (less than 40 cm DBH) (figure 2c,f) and were pronounced (e.g. there was an approximately 30% decrease in trees less than 20 cm DBH), with mortality effects decreasing moving from small to medium size classes. The losses of small stems (DBH < 30 cm) increased over time (2015–2018), but this tendency was not detected in larger stems (DBH ≥ 30 cm). Losses in unburned plots, however, were smaller and more evenly distributed among stem sizes (less than 15% in all size classes) (figure 2b,d).

Decreases in both stem density and AGB were found in all three growth forms in the burned plots (electronic supplementary material, table S2) and were particularly large in lianas, which lost 38.6% in stem density and 38.1% in AGB. Trees and palms lost, respectively, 28.0 and 14.6% in stem density, and 12.1 and 27.2% in AGB. These same comparisons for the unburned plots showed much smaller losses or no change trends: trees, palms and lianas lost, respectively, 6.1, 7.4 and 11.3% in stem density, and only lianas lost AGB (12.6%).

## (c) Changes in forest dynamics

Recruitment and mortality were the main drivers of post-fire AGB dynamics (figure 3). During the first and second intervals of measurement, recruitment rates were similar between burned and unburned plots, but during the third interval, the rate decreased in burned plots, reaching values 2.7 times lower than in unburned plots ($p < 0.05$). The number of tree recruits in the third survey interval was

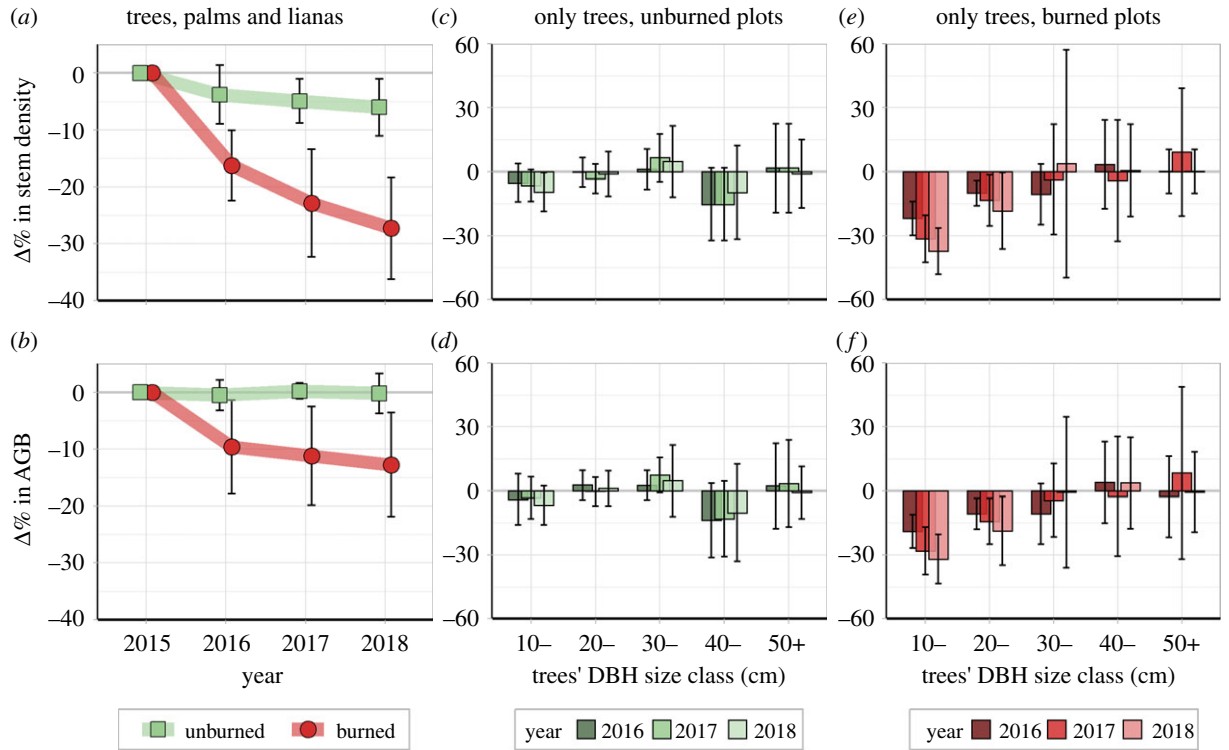

**Figure 2.** Percentage of change (Δ%) in stem density and AGB per year and DBH size class in unburned and burned plots over 3 years after the forest fire. Symbols and vertical bars represent, respectively, mean and one standard deviation (±s.d.) between plots. (Online version in colour.)

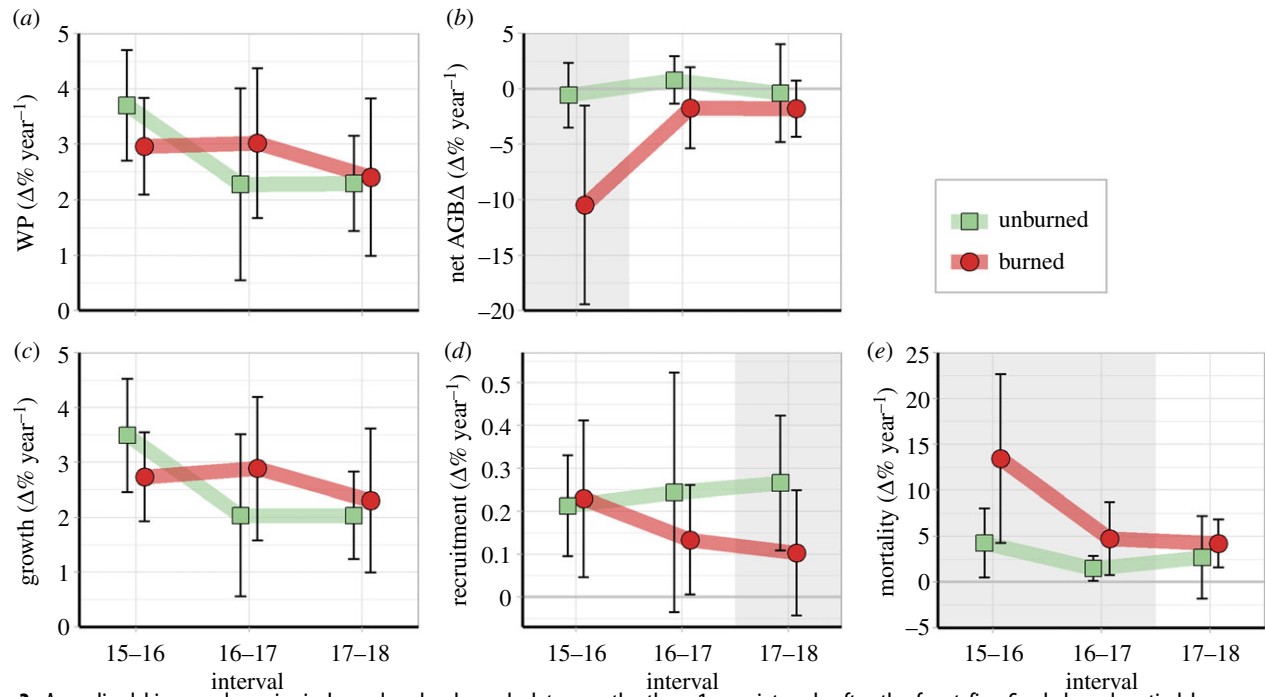

**Figure 3.** Annualized biomass dynamics in burned and unburned plots over the three 1-year intervals after the forest fire. Symbols and vertical bars represent, respectively, mean and one standard deviation (±s.d.) between plots. Significance differences ($p < 0.05$) between unburned and burned plots are indicated by the grey background. (Online version in colour.)

$3 \pm 4$ stems ha$^{-1}$ y$^{-1}$ in the burned area and $8 \pm 3$ stems ha$^{-1}$ y$^{-1}$ in the unburned area. Mortality in burned plots was approximately 3 times higher than in unburned plots in the first and second intervals only ($p < 0.05$), reaching $31.8 \pm 33.2$ and $8.3 \pm 5.5$ Mg ha$^{-1}$ y$^{-1}$, respectively. WP displayed this same behaviour due to the low contribution of recruitment to the overall AGB. Net AGBΔ of burned plots in the first interval ($-10.5 \pm 9.0\%$ y$^{-1}$) was 18.4 times lower than in unburned plots, but there was no difference in the other intervals ($p > 0.05$).

AGB absolute rates (Mg ha$^{-1}$ year$^{-1}$) are available in electronic supplementary material, figure S4.

### (d) Changes in stem-level growth
While CH had no significant effect on radial growth and carbon accumulation ($p > 0.05$), WD had a significant negative effect on radial growth ($p < 0.05$), and DBH had a significant positive effect on carbon accumulation ($p < 0.001$) (figure 4). These relationships indicate,

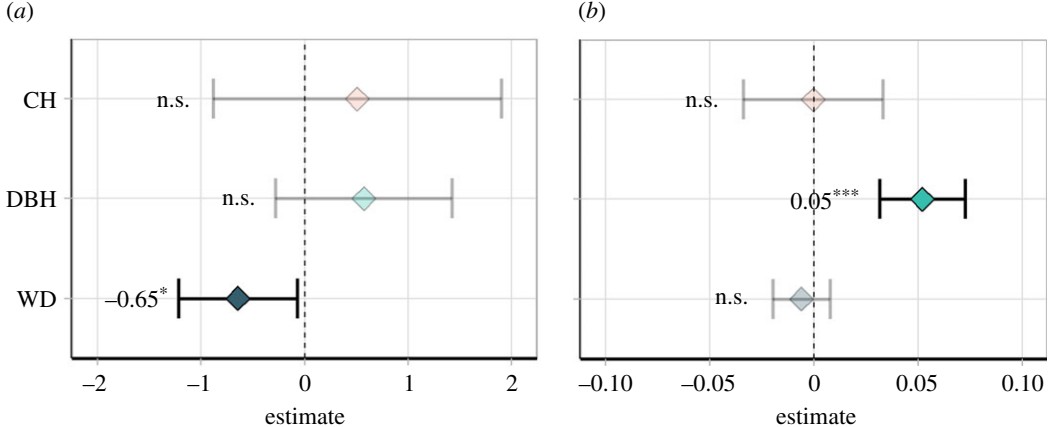

**Figure 4.** Estimates from linear models adjusted to predict the difference in radial growth (*a*) and carbon accumulation (*b*) between similar trees in burned and unburned plots according to CH, DBH and WD. Bars represent 95% confidence intervals. Transparent symbols indicate non-significant estimates ($p > 0.05$). (Online version in colour.)

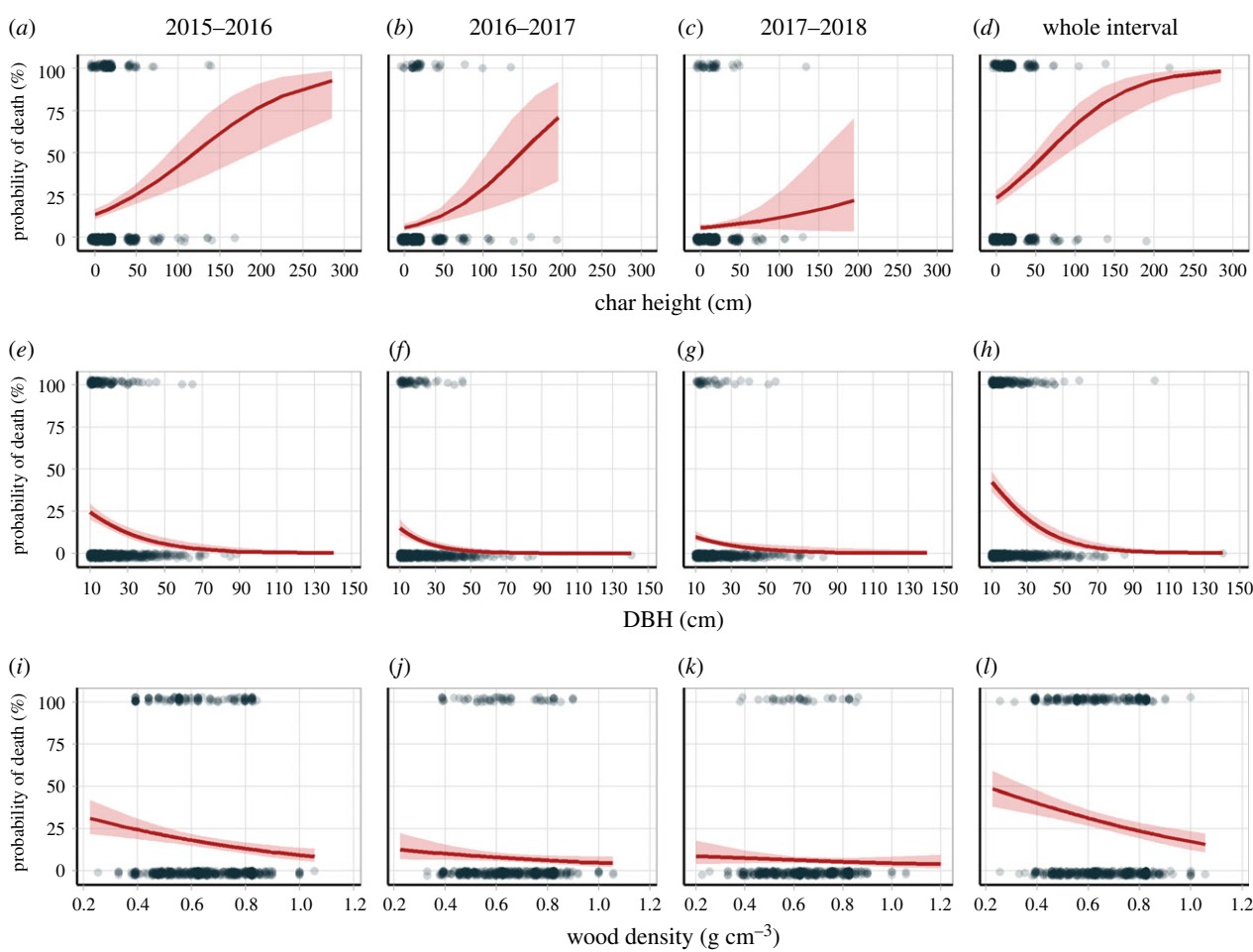

**Figure 5.** Tree mortality probability over time as a function of CH (*a–d*), DBH (*e–h*) and WD (*i–l*). The central line is the predicted values of each variable's response (±95% confidence interval). Dots represent raw data. (Online version in colour.)

respectively, that the lower the WD of a tree, the greater the increase in radial growth in burned plots up to 3 years after the fire, and the greater the tree size, the greater the increase in carbon accumulation in burned plots.

## (e) Changes in mortality probability

Average-DBH and average-WD trees were more than 95% likely to die within 2 years if affected by an intense fire (CH > 200 cm, figure 5*d*). These high-CH trees died within

the first year of the study and thus do not appear in the subsequent years' predictions (figure 5*b,c*). However, for those trees surviving the first post-fire year, trees with approximately 200 cm CH had a 71% likelihood of mortality in the second interval (figure 5*b*). Small trees (e.g. 10 cm DBH) and large trees (e.g. 110 cm DBH) had, respectively, 43% and 0% probabilities of mortality within 3 years after fire (figure 5*h*). Moreover, low-WD trees (approx. 0.2 g cm³) and high-WD trees (approx. 1.0 g cm³) had, respectively, 49%

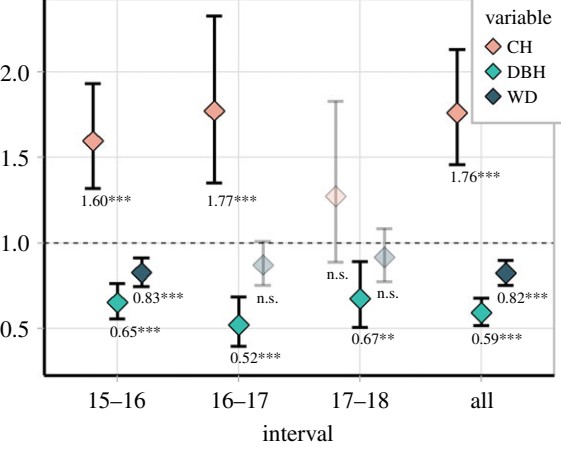

**Figure 6.** Multiplicative factor (odds ratio) for predicting the change in tree mortality probability following the fire, as a function of CH, DBH and WD. Points and bars represent mean and 95% confidence intervals, respectively. Bars represent 95% confidence intervals. Transparent symbols indicate non-significant estimates ($p > 0.05$). This figure should be interpreted as: for each 30 cm increase in CH, 10 cm in DBH and 0.1 g cm$^{-3}$ in WD, the inherent odds of a tree to die after the fire is altered by the multiplicative factor on the x-axis. (Online version in colour.)

and 17% probabilities of mortality (figure 5*l*). In summary, CH showed a positive relationship and DBH and WD showed negative relationships with the probability of death. However, CH and DBH presented saturation points—with maximum and minimum probabilities of death for approximately 2 m CH and 70 cm DBH, respectively. By contrast, the negative WD relationship was more linear, without saturation.

A comparison of all predictors shows that CH had a higher effect on tree mortality than DBH and WD (figure 6). Analysing the entire 3-year interval, each 30 cm increase in CH increased the odds of a tree dying by a factor of 1.76 (or 76%, $p < 0.001$), while each 10 cm in DBH and 0.1 g cm$^{-3}$ in WD decreased the odds of a tree dying by 0.59 (or −41%, $p < 0.001$) and 0.82 (or −18%, $p < 0.001$), respectively. Therefore, the effect of a 30 cm increase in CH on increasing mortality probability was, on average, 1.8 and 4.2 greater than the counterbalancing opposite effect of a 10 cm increase in DBH and a 0.1 g cm$^{-3}$ increase in WD, respectively. Analysing annual intervals, CH's effect was significant ($p < 0.05$) and greater than the other variables' effects until 2 years after the fire.

These significant effects reflect the best models selected for each interval through the preliminary ranking ($\Delta$AICc < 1.3; electronic supplementary material, table S3). When we repeated the model selection for unburned and burned plots with DBH and WD as predictors, only WD had significant effects on tree mortality in unburned plots considering all analysed intervals ($\Delta$AICc < 0.7; electronic supplementary material, table S4). The same modelling approach (with CH and DBH) for palms in burned plots showed no significant effects for all variables in all intervals ($\Delta$AIC ≤ 1.1; electronic supplementary material, table S5).

## 4. Discussion

### (a) The heterogeneity of post-fire changes in forest structure and biomass

Our forest site was affected by an understorey fire of low-intensity and irregular spread that did not directly affect all

trees in burned plots—a pattern that was clear from our data on tree-level fire mark. These forest fires caused structural losses of moderate magnitude and high spatial variability. The stem density loss (overall tree mortality rate) reported in this study (28 ± 8%) is similar to those reported for a similar time interval in a previous study also conducted in the Purus-Madeira northern region but in a less fragmented landscape (16 ± 16%; electronic supplementary material, table S6) [29]. This loss is less extreme than those reported for similar time intervals in more seasonal regions in the central-eastern (36–66%) [15,17,18,50] and in the southwestern Amazon (approx. 50%) [11,51]. However, our results are more similar to those found in the phytogeographic borders of the Amazon (less than or equal to 23%) [9,15,52]. The reasons for the lower impacts of fire on biomass in the less seasonal region we studied and the phytogeographic borders of the Amazon likely diverge: trees in highly seasonal phytogeographic border regions tend to have adaptive protections against fire, in particular thicker bark [23], while in less seasonal regions higher surface fuel moisture limits fire spread and intensity even during an extreme drought such as the 2015/2016 El Niño period. In spite of this higher margin of moisture-derived resi lowering fire intensity in less seasonal Amazon, these areas demand strict protection owing to their high carbon storage and biodiversity [53].

We also reported losses in the smaller tree size classes, agreeing with other studies analysing changes in the 3 years following the fire [17,19,29]. However, we did not find increased mortality of large trees, in contrast with previous studies [19,29,54]. While losses of 30–48% in tree stems greater than or equal to 50 cm in DBH up to 3 or 4 years after fire were reported in these studies, we found a near-zero average change in this DBH class (0.2 ± 10.2%). However, if a delayed post-fire mortality of large stems can continue for up to 8 years [20], a decline of large-diameter trees might still happen in our study area.

Differences among fire impacts found in this and other studies likely result from differences in pre-fire vegetation characteristics, local drought severity and fire behaviour [9,11,50]. Furthermore, several methodological differences may reduce comparability. First, most studies used single census measurement and assessed fire impacts by comparing burned and near-by unburned forests. If we had a single census only in any year from 2015 to 2018 for example, we would have found greater biomass in the burned areas compared to the unburned areas and thus we would not be able to report the actual AGB losses in burned forests. This study and a few others are unique in temporarily tracking changes within the same plots [9,15,17,20]. Second, sampling burned forests is complex. In natural experiments, like ours, a random or systematic plot placement is sometimes limited due to the irregular fire area and access restrictions in private lands [15]. As a consequence, burned forests are underrepresented in natural experiments, and studies may fail to capture the whole spatial variability of fire degradation. Therefore, it is critical that remote sensing technologies such as LiDAR (light detection and ranging) should be employed complementarily alongside traditional field inventories [55,56] to address the spatial heterogeneity of fire impacts on forests.

### (b) Fire-mediated changes in tree mortality and recruitment

We found that fire initially increased tree mortality rates, while wood production (growth and recruitment) did not

counterbalance the fire impact—as may have been predicted from decreasing competition and/or fire-mediated nutrient mobilization. Thus, our findings do not agree with prior investigations that observed low-intensity fires enhance forest growth rates to partially offset carbon emissions from enhanced mortality [20,24]. Nevertheless, we found persistent reduced recruitment among the burned plots, a pattern that has never been reported before for the Amazon forests. We hypothesize that this lower recruitment is due to a lack of saplings (DBH < 10 cm, not tracked in our study)—our observations in the field suggested that fire killed a generation of saplings, which is consistent with prior findings (e.g. 76% sapling mortality) [14]. However, this decrease in tree recruitment is likely to be accompanied by increasing seedling recruitment that may reach 10 cm DBH in the following post-fire intervals. Alternatively, the enhancement of two native herbaceous bamboo species registered in some of our fire-affected plots (electronic supplementary material, figure S2) [57] may further reduce recruitment through competitive suppression of saplings. Finally, our observations of temporal changes in mortality rates highlight the importance of multi-temporal censuses starting at the earliest time possible. For example, if we had started our censuses in 2017 (2 years after the fire), we would not have registered the markedly higher post-fire mortality.

Continued monitoring of these plots in regular intervals (annual or biannual) is important to improve our understanding of the related carbon fluxes (emissions and uptake) [58], the recovery time to pre-fire states [20] and/or eventual disruption of carbon dynamics by tree mortality (e.g. caused by additional drought and fire events) [59].

## (c) Fire intensity and stem morphology effects on growth and mortality

We found that low-WD and large-sized trees grew more in the burned area, complementing the only previous study analysing the factors affecting post-fire growth in the Amazon, which found a similar WD effect and indicated that low-WD trees were likely to be benefitting from a large fire-mediated pulse of nutrients [24]. This same study did not find significant effects of the change in live basal area surrounding each surviving tree. However, since we did not investigate this effect and our sample size was much greater than that of [24] (575 × 64 pairs), we do not discard the hypothesis of the surviving large trees being benefitted from reduced competitive stress after the fire has reduced small tree densities. In this case, the fire could act similarly to tree thinning, a common silvicultural technique for improving individual tree growth [60]. This hypothesis should be investigated in future studies. Additionally, the greater growth of large trees in burned forests must be considered with caution, because large trees are susceptible to delayed mortality, as previously mentioned.

Fire intensity, proxied by the CH on the tree trunk, did not affect tree growth but had intense and transitory effects on its likelihood to die, surpassing the protective effects of higher WD and larger tree sizes. We found that fire intensity and tree size are important to explain immediate mortality (up to two years) while WD may be an important predictor in a longer interval. Prior studies have already shown the effects of these variables on post-fire tree mortality in tropical forests [21,22]; however, our results quantitatively

characterize how trees with certain resilient plant traits (greater sizes and WD) still may not be able to survive under high-intensity fires. Since fire intensity is linked to drought severity [9], forest fire impact and extent in the less seasonal Amazon may increase in the future with widespread anthropogenic disturbances degrading forest canopies and creating hotter drier understorey conditions and with more widespread and intense droughts forecasted as a result of climate change [2,59].

## (d) Insights for future works and forest protection strategies

Further studies should focus on long-term post-fire monitoring to investigate if the reported delayed large tree mortality occurs on broad scales in the Amazon, to improve our understanding of the factors driving post-fire forest recovery (growth and, mainly, recruitment) and to improve estimates of recovery time to pre-fire structural states [20]. Specifically, we recommend that further studies should investigate forest fire intensity and severity in less seasonal regions of the Amazon, placing fire-affected permanent plots in regions of 0–1 dry season months.

To support future decision making to avoid large-scale forest fire in the Amazon, we also recommend the development of two products: (i) mapping of the forest fire risk, based on the spatial variability of the drivers of fire spread and intensity, such as the rainfall regime, amount of surface fuel and forest microclimate [2,9,26]; and (ii) mapping of potential fire impact (fire sensitivity/resistance), derived mainly from the spatiality of morphological plant traits [21–23]. These products could indicate specific forest types and regions demanding special attention regarding fire occurrence, such as forests recovering from disturbances, which are susceptible to even greater losses if affected by a second fire event [11,15]; and seasonally flooded forests, which are highly susceptible and sensitive to fire due to their higher fine-fuel load, and flammable root mat [29]. As previously cited, one critical region under threat of fire degradation is the Purus-Madeira, which is fully permeated by flooded forest networks. Without proper land-use regulation, the current intention of Brazil's government to pave the BR-319 road will increase deforestation in the Purus-Madeira [61], increasing ignition sources and the associated risk of large-scale forest dieback in this region.

Data accessibility. Raw forest inventory data: ForestPlots.net (codes NOC_01 to NOC_10 and TIC_01 to TIC_08). Processed data and data analysis are available from the Dryad Digital Repository: https://doi.org/10.5061/dryad.ncjsxkstf [62]. Assembly of works on burned Amazonian forests uploaded as online supplementary material.

Authors' contributions. A.P.-L: conceptualization, data curation, formal analysis, methodology, and writing-original draft; C.V.J.S.: conceptualization, data curation, methodology, writing-review and editing; J.B.: methodology, supervision, writing-review and editing; L.M.R.: data curation, writing-review and editing; W.A.C.: formal analysis, writing-review and editing; C.A.N.: formal analysis, writing-review and editing; C.T.A.: data curation, writing-review and editing; C.H.L.S.J.: data curation, writing-review and editing; H.L.G.C.: data curation, writing-review and editing; R.D.: formal analysis, writing-review and editing; S.C.S.: formal analysis, writing-review and editing; P.M.G.: conceptualization, funding acquisition, methodology, resources, writing-review and editing; L.E.O.C.A.: conceptualization, funding acquisition, methodology, resources, supervision, writing-review and editing

All authors gave final approval for publication and agreed to be held accountable for the work performed therein.

**Competing interests.** We declare we have no competing interests.

**Funding.** This work was part of A.P.-L.'s PhD thesis, funded by Fundação de Amparo à Pesquisa do Estado de São Paulo (FAPESP grant no. 2016/21043-8). This work was also funded by Conselho Nacional de Desenvolvimento Científico e Tecnológico (CNPq 458022/2013-6), Coordenação de Aperfeiçoamento de Pessoal de Nível Superior (CAPES, code 001) and Project Environmental Satellite Monitoring in the Amazon Biome (MSA-BNDES)–Activity 7 (Amazon Fund 14209291). C.V.J.S. was funded by Research England (QR Strategic Priorities Fund); W.A.C. was funded by CNPq 140261/2018-4; C.T.A. was funded by CNPq 140502/2016-5 and FAPESP 2020/06734-0;

C.H.L.S.J. was funded by CAPES (Code 001); H.L.G.C. and R.D. were funded by FAPESP (2018/14423-4 and 2019/21662-8). S.C.S. was funded by USDA NIFA and National Science Foundation (NSF) (DEB-1950080 and 1754357).

**Acknowledgements.** We thank INPA's Agroecosystem Laboratory, INCT-SERVAMB and INCT-CENBAM for logistical support in the field, and the ForestPlots.net team for the support in standardizing the datasets. We are also grateful to our parabotanists Antônio Mello, José Souza and Izaias Brasil; all field assistants; all landowners and the local community for supporting our field surveys.

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
