## [Peer Review File · Proceedings of the Royal Society B: Biological Sciences]

Review History

RSPB-2021-0094.R0 (Original submission)

Review form: Reviewer 1

Recommendation

Accept with minor revision (please list in comments)

Scientific importance: Is the manuscript an original and important contribution to its field?

Good

General interest: Is the paper of sufficient general interest?

Good

Quality of the paper: Is the overall quality of the paper suitable?

Good

Is the length of the paper justified?

Yes

Should the paper be seen by a specialist statistical reviewer?

No

Do you have any concerns about statistical analyses in this paper? If so, please specify them explicitly in your report.

No

It is a condition of publication that authors make their supporting data, code and materials available - either as supplementary material or hosted in an external repository. Please rate, if applicable, the supporting data on the following criteria.

Is it accessible?

Yes

Is it clear?

Yes

Is it adequate?

Yes

Do you have any ethical concerns with this paper?

No

Comments to the Author

I believe this is a robust analysis that provides insights into the effects of fire in an area that should rarely be affected by wildfire. Overall, the writing is good but there are a few places that require attention (see below).

Obviously, the best study design are permanent plots that get burnt. That the study is a natural experiment needs to be more clearly explained at the outset, including the assumptions that the unburnt sites are similar to the burnt sites. Another assumption is how representative the fires are: they occur in a fragmented forest and humans attempted to control them.

The study makes repeated measurements a virtue but why this is important is not clearly explained. Also, if repeated measures are important to detected legacy effects of fire then what is the recommendation of the ideal temporal sampling design for future studies? Why not just have single sample after several years compared to adjacent plots?

Figure S1 'live coal from a burning wood log' this is clumsy, do you mean 'smouldering wood and charcoal'

Line 159 'Mean wood density (WD) was retrieved for each individual from the Global Wood Density Database [41], following this priority order: species (42% of all individuals), genus (50%), family (2%), and plot mean (6%), being this last applied to individuals without taxonomic information.' I am struggling to understand this, what is a 'priority order'.

Review form: Reviewer 2

Recommendation

Accept with minor revision (please list in comments)

Scientific importance: Is the manuscript an original and important contribution to its field?

Good

General interest: Is the paper of sufficient general interest?

Good

Quality of the paper: Is the overall quality of the paper suitable?

Good

Is the length of the paper justified?

Yes

Should the paper be seen by a specialist statistical reviewer?

No

Do you have any concerns about statistical analyses in this paper? If so, please specify them explicitly in your report.

No

It is a condition of publication that authors make their supporting data, code and materials available - either as supplementary material or hosted in an external repository. Please rate, if applicable, the supporting data on the following criteria.

Is it accessible?

Yes

Is it clear?

Yes

Is it adequate?

Yes

Do you have any ethical concerns with this paper?

No

Comments to the Author

This empirical paper does a good job of documenting the impacts of fire on tree dynamics and structure in the less seasonal part of the Amazon. The paper is clear and well-written. It provides important insights on the insights of fire in a less studied region.

Abstract

- I would suggest indicating that only trees > 10 cm dbh were studied in the abstract

Introduction

- No major comments. It is clear and well written. I like the paragraph that clearly identified the gaps the study is addressing.

Methods:

- Why was an unbalanced study design of 6 unburned and 12 burned plots decided on?
- What is the justification for choosing plots of 0.25 ha as well as deciding on long linear transects? It would be good to put some justification into the text.
- I have some concerns about how recruitment was measured. Since only trees > 10 cm dbh were censused, recruitment may be better thought of as in-growth. If the authors had installed seedling plots that would have been a better measure of recruitment. I recommend de-emphasizing this result or reframing it.
- I like the paired individual tree analysis.

Results:

- The results could benefit from more details on burn severity. Could the authors create a figure that shows how it varied across the transects? As written now it is hard to understand the spatial extent of the burn patterns.

Discussion:

- No suggestions.

Decision letter (RSPB-2021-0094.R0)

15-Apr-2021

Dear Mrs Pontes-Lopes

I am pleased to inform you that your manuscript RSPB-2021-0094 entitled "Drought-driven wildfire impacts on structure and dynamics in a wet Central Amazonian forest" has been accepted for publication in Proceedings B. Congratulations!!

The referee(s) have recommended publication, but also suggest some minor revisions to your manuscript. Therefore, I invite you to respond to the referee(s)' comments and revise your manuscript. Because the schedule for publication is very tight, it is a condition of publication that you submit the revised version of your manuscript within 7 days. If you do not think you will be able to meet this date please let us know.

Online supplementary material will also carry the title and description provided during submission, so please ensure these are accurate and informative. Note that the Royal Society will not edit or typeset supplementary material and it will be hosted as provided. Please ensure that

the supplementary material includes the paper details (authors, title, journal name, article DOI). Your article DOI will be 10.1098/rspb.[paper ID in form xxxx.xxxx e.g. 10.1098/rspb.2016.0049].

It is a condition of publication that data supporting your paper are made available either in the electronic supplementary material or through an appropriate repository. Please see our Data Sharing Policies <https://royalsociety.org/journals/authors/author-guidelines/#data>.

Sincerely,

Dr John Hutchinson, Editor

Associate Editor

Comments to Author:

This MS has been very well received by the reviewers, who have asked for a very limited set of additional details to be added and clarifications to be made. It's a nicely constructed field study with a strong dataset that's been well analysed and well presented.

Reviewer(s)' Comments to Author:

Referee: 1

Comments to the Author(s)

I believe this is a robust analysis that provides insights into the effects of fire in an area that should rarely be affected by wildfire. Overall, the writing is good but there are a few places that require attention (see below).

Obviously, the best study design are permanent plots that get burnt. That the study is a natural experiment needs to be more clearly explained at the outset, including the assumptions that the unburnt sites are similar to the burnt sites. Another assumption is how representative the fires are: they occur in a fragmented forest and humans attempted to control them.

The study makes repeated measurements a virtue but why this is important is not clearly explained. Also, if repeated measures are important to detected legacy effects of fire then what is the recommendation of the ideal temporal sampling design for future studies? Why not just have single sample after several years compared to adjacent plots?

Figure S1 'live coal from a burning wood log' this is clumsy, do you mean 'smouldering wood and charcoal'

Line 159 'Mean wood density (WD) was retrieved for each individual from the Global Wood Density Database [41], following this priority order: species (42% of all individuals), genus (50%), family (2%), and plot mean (6%), being this last applied to individuals without taxonomic information.' I am struggling to understand this, what is a 'priority order'.

Referee: 2

Comments to the Author(s)

This empirical paper does a good job of documenting the impacts of fire on tree dynamics and structure in the less seasonal part of the Amazon. The paper is clear and well-written. It provides important insights on the insights of fire in a less studied region.

Abstract

- I would suggest indicating that only trees > 10 cm dbh were studied in the abstract

Introduction

- No major comments. It is clear and well written. I like the paragraph that clearly identified the gaps the study is addressing.

Methods:

- Why was an unbalanced study design of 6 unburned and 12 burned plots decided on?
- What is the justification for choosing plots of 0.25 ha as well as deciding on long linear transects? It would be good to put some justification into the text.
- I have some concerns about how recruitment was measured. Since only trees > 10 cm dbh were censused, recruitment may be better thought of as in-growth. If the authors had installed seedling plots that would have been a better measure of recruitment. I recommend de-emphasizing this result or reframing it.
- I like the paired individual tree analysis.

Results:

- The results could benefit from more details on burn severity. Could the authors create a figure that shows how it varied across the transects? As written now it is hard to understand the spatial extent of the burn patterns.

Discussion:

- No suggestions.

Author's Response to Decision Letter for (RSPB-2021-0094.R0)

See Appendix A.

Decision letter (RSPB-2021-0094.R1)

23-Apr-2021

Dear Mrs Pontes-Lopes

I am pleased to inform you that your manuscript entitled "Drought-driven wildfire impacts on structure and dynamics in a wet Central Amazonian forest" has been accepted for publication in Proceedings B.

If you are likely to be away from e-mail contact please let us know. Due to rapid publication and an extremely tight schedule, if comments are not received, we may publish the paper as it stands. If you have any queries regarding the production of your final article or the publication date please contact procb_proofs@royalsociety.org

Data Accessibility section

Open Access

Paper charges

Sincerely,

Appendix A

Response to Reviewers on the Ms. Ref. N°. RSPB-2021-0094 by Pontes-Lopes et al.

Manuscript entitled: Drought-driven wildfire impacts on structure and dynamics in a wet Central Amazonian forest

22 April 2021

Dear Dr. John Hutchinson,

We are pleased that the *Proceedings B* and both reviewers saw the importance of our work. The valuable suggestions certainly improved the quality of our manuscript. After considering all the suggestions, the article has undergone some minor changes, summarized as follows:

- Some punctual clarifications were made in the abstract, introduction, and results section;
- The methodology was expanded to highlight that our study was a natural experiment, to state our assumption of representativeness; and to expose the reason for elongated plots;
- The discussion was expanded to complement the interpretation of the decreased recruitment, and to state the importance of repeated measurements.

The changes in the manuscript are detailed below (in blue), point-by-point, for each reviewer's comments. Additionally, we have submitted the revised manuscript with tracked changes. We hope that these changes are in agreement for publication in the *Proceedings B*. Please, let us know if further clarification is needed.

Best regards,
Aline Pontes Lopes
Corresponding Author

--- Comments to Author:

This MS has been very well received by the reviewers, who have asked for a very limited set of additional details to be added and clarifications to be made. It's a nicely constructed field study with a strong dataset that's been well analysed and well presented.

[Response R1] We really appreciate this comment and the valuation of our efforts.

Reviewers' Comments to Author:

--- Referee: 1

I believe this is a robust analysis that provides insights into the effects of fire in an area that should rarely be affected by wildfire. Overall, the writing is good but there are a few places that require attention (see below).

[R2] We appreciate the positive assessment and the valuable suggestions, that helped to improve the quality of this manuscript.

Obviously, the best study design are permanent plots that get burnt. That the study is a natural experiment needs to be more clearly explained at the outset, including the assumptions that the unburnt sites are similar to the burnt sites. Another assumption is how representative the fires are: they occur in a fragmented forest and humans attempted to control them.

[R3] As the reviewer has noted, we used the expression “natural experiment” in the discussions only (lines 394 and 396, of the manuscript with tracked changes). We prefer not to use this expression at the outset to avoid misinterpretations, once that some readers not familiarized with this expression may understand that this means natural wildfires – which do not occur in the Amazon. Therefore, we added the expression “*uncontrolled forest fires*” at the end of the introduction (line 97) and added the following sentence in the methods (line 153) to emphasized that this was not a burn experiment:

“We were able to set some of our plots during the wildfires. However, this was not a burn experiment, and we did not set and control the fire.”

We understand that the first assumption – that the unburnt sites are similar to the burnt sites – is an indirect assumption of our study, because we compared the temporal changes that happened in both plot types (unburned *versus* burned). However, we prefer not to present this assumption in the manuscript for two reasons. Firstly, this assumption may lead some readers to misunderstand that our baseline was derived from the unburned plot's average (as in single-time assessments). Secondly, our data showed that this assumption was not completely valid. The burned plots' average biomass was greater compared to the unburned plots in all censuses, especially in the first census (Table S2).

Finally, we totally agree with the other assumption – of how representative the fires are – and added the following sentence to the manuscript (lines 155-159), together with a new reference (Armenteras *et al.* 2017).

“Furthermore, we consider that the surveyed burned forest area, composed by natural and human-made borders (figure 1d), is regionally representative once that wildfires do not naturally occur in this wet environment, depending on ignition vectors associated to human presence and climatological conditions [40].”

However, note that our results may also represent more preserved and isolated forests in the Purus-Madeira region, as shown by the statistical comparisons between our burned plots and Resende et al. burned plots (lines 363-366 and table S6). To clarify the importance of this sentence, we have added the following expression to it: *“but in a less-fragmented landscape”* (line 365). Furthermore, we are not aware of any governmental effort (either federal or local) of controlling these forest fires. During our first field campaigns we witnessed residents from only one private property trying to make fire breaks to impede the fire from reaching their forest area, but they didn't succeed (figure S1, top left).

The study makes repeated measurements a virtue but why this is important is not clearly explained. Also, if repeated measures are important to detected legacy effects of fire then what is the recommendation of the ideal temporal sampling design for future studies? Why not just have single sample after several years compared to adjacent plots?

[R4] To clarify the importance of repeated measurements, we have added the following sentence to the discussions (line 423-426).

“Continued monitoring of these plots in regular intervals (annual or biannual) is important to improve our understanding of the related carbon fluxes (emissions and uptake) [58], the recovery time to pre-fire states [20], and/or eventual disruption of carbon dynamics by tree mortality (e.g. caused by additional drought and fire events) [59].”

We consider that the present study does not have a time series long enough to recommend an ideal temporal sampling design. However, any recommendation depends on the study objectives. To model carbon fluxes, we would suggest one survey immediately after fire (or ideally before the fire) to define a baseline and then yearly or biannual measurements up to ~8 years, when delayed mortality events may have ceased, and only regeneration processes may have remained.

Figure S1 'live coal from a burning wood log' this is clumsy, do you mean 'smouldering wood and charcoal'

[R5] Thanks for the attention. We have changed the text to match the suggested expression.

Line 159 'Mean wood density (WD) was retrieved for each individual from the Global Wood Density Database [41], following this priority order: species (42% of all individuals), genus (50%), family (2%), and plot mean (6%), being this last applied to individuals without taxonomic information.' I am struggling to understand this, what is a 'priority order'.

[R6] We've modified this sentence for clarification as following:

“Mean wood density (WD) was retrieved for each individual from the Global Wood Density Database [41] according to its most detailed taxonomic identification [42], being 42% of all individuals classified at the species level, 50% up to the genus level only, and 2% at the family level only. The remaining 6% were not identified and received plot-level mean WD values.”

--- Referee: 2

Comments to the Author(s)

This empirical paper does a good job of documenting the impacts of fire on tree dynamics and structure in the less seasonal part of the Amazon. The paper is clear and well-written. It provides important insights on the insights of fire in a less studied region.

[R7] We really appreciate this comment and all the constructive suggestions.

Abstract

- I would suggest indicating that only trees > 10 cm dbh were studied in the abstract.

[R8] We've added this information to the abstract.

Introduction

No major comments. It is clear and well written. I like the paragraph that clearly identified the gaps the study is addressing.

[R9] We acknowledge these positive feedbacks.

Methods:

Why was an unbalanced study design of 6 unburned and 12 burned plots decided on?

[R10] As we expect that unburned forests present the less structural variations than the fire-affected forests, it is usual to adopt a smaller number of unburned plots (e.g., Berenguer et al. 2014). These different structural variations are exposed by the unburned plots' standard deviation in 2015 ($\pm 36.9 \text{ Mg ha}^{-1}$), which is 0.6 times lower than the burned plots' standard deviation ($\pm 63.9 \text{ Mg ha}^{-1}$). These values can be calculated from the plot-based aboveground biomass values (TAGB_ha column, filtered for 2015) available in the Dryad repository ('Pontes-Lopes-et-al_PRS-B_Q2_Structure_Data.xlsx' spreadsheet).

What is the justification for choosing plots of 0.25 ha as well as deciding on long linear transects? It would be good to put some justification into the text.

[R11] As suggested, we've added the following sentence to the methods (line 163): *“Despite not randomly placed, these rectangular plots minimize potential selection bias [15].”* Moreover, quarter-hectare is a usual sample size in Amazonian studies (Haugaasen et al. 2003, Barlow & Peres 2004, Berenguer et al. 2014), being considered as the minimum sample size required for sampling tree aboveground biomass in this region (Keller et al. 2001).

I have some concerns about how recruitment was measured. Since only trees > 10 cm dbh were censused, recruitment may be better thought of as in-growth. If the authors had installed seedling plots that would have been a better measure of recruitment. I recommend de-emphasizing this result or reframing it.

[R12] We totally understand these concerns. While we have detected decreased tree recruitment (in-growth in our smallest tree size class) owing to the fire, there may have been increasing seedling recruitment. Therefore, we have improved our definition of recruitment (line 199) and added the following sentence in the discussions (lines 413-415).

“However, this decrease in tree recruitment is likely to be accompanied by increasing seedling recruitment, that may reach 10 cm DBH in the following post-fire intervals.”

I like the paired individual tree analysis.

[R13] We appreciate this positive feedback.

Results:

- The results could benefit from more details on burn severity. Could the authors create a figure that shows how it varied across the transects? As written now it is hard to understand the spatial extent of the burn patterns.

[R14] Following this suggestion, we have inserted a new Figure S3 in the Supplementary Material. This new figure clearly exposes how the burn patterns vary between and within plots.

Discussion:

- No suggestions.

References presented in this “Response to referees”

Armenteras, D., Barreto, J. S., Tabor, K., Molowny-Horas, R., & Retana, J. (2017). Changing patterns of fire occurrence in proximity to forest edges, roads and rivers between NW Amazonian countries. *Biogeosciences*, 14(11), 2755–2765. <https://doi.org/10.5194/bg-14-2755-2017>

Barlow, J., & Peres, C. A. (2004). Ecological responses to El Niño–induced surface fires in central Brazilian Amazonia: management implications for flammable tropical forests. *Philosophical Transactions of the Royal Society of London. Series B: Biological Sciences*, 359(1443), 367–380. <https://doi.org/10.1098/rstb.2003.1423>

Berenguer, E., Ferreira, J., Gardner, T. A., Aragão, L. E. O. C., De Camargo, P. B., Cerri, C. E., Durigan, M., Oliveira, R. C. De, Vieira, I. C. G., & Barlow, J. (2014). A large-scale field assessment of carbon stocks in human-modified tropical forests. *Global Change Biology*, 20(12), 3713–3726. <https://doi.org/10.1111/gcb.12627>

Haugaasen, T., Barlow, J., & Peres, C. A. (2003). Surface wildfires in central Amazonia : Short-term impact on forest structure and carbon loss. *Forest Ecology and Management*, 179(1–3), 321–331. [https://doi.org/10.1016/S0378-1127\(02\)00548-0](https://doi.org/10.1016/S0378-1127(02)00548-0)